# The Relationship between Dynamic and Static Deformation Modulus of Unbound Pavement Materials Used for Their Quality Control Methodology

**DOI:** 10.3390/ma15082922

**Published:** 2022-04-16

**Authors:** Martin Decký, Marian Drusa, Daniel Papán, Juraj Šrámek

**Affiliations:** 1Department of Highways Engineering, University of Zilina, Univerzitna 8215/1, 010 26 Zilina, Slovakia; martin.decky@uniza.sk; 2Department of Geotechnics, University of Zilina, Univerzitna 8215/1, 010 26 Zilina, Slovakia; marian.drusa@uniza.sk; 3Department of Structural Mechanics and Applied Mathematics, University of Zilina, Univerzitna 8215/1, 010 26 Zilina, Slovakia; 4Department of Technology and Construction Management, University of Zilina, Univerzitna 8215/1, 010 26 Zilina, Slovakia; juraj.sramek@uniza.sk

**Keywords:** deformation characteristics, correlations, dynamic force, earthworks quality control, static theory of impact

## Abstract

In the present study, credible analytical and numerical models are developed in order to explain the apparent discrepancies in the ratios of static and dynamic deformation models for assessing the quality of mechanical efficiency of transport structures in Central Europe. Through of experience, authors specifically deal with the comparison of two commonly used methods: the dynamic load plate test, known as the lightweight dynamic test and the static plate load test. This paper presents the relevant correlation dependency of the most commonly used quantification characteristics in earthworks quality control. Their correlation was obtained by applying the static theory of impact to earthworks quality control, which allows for the application of several quality control methods, in line with other member states of the European Union, specifically with regard to constructions under various boundary conditions (climate, soil moisture of the specified layer). According to an analysis of the results of comparisons of static and dynamic load tests, analytical and numerical models of the subsoil formed by soils and uncemented structural materials, respectively, the linear calculation usually used in the conditions of Central Europe does not have universal validity. Rather than relying on the analytical and FEM models for the soil, the authors have determined that the above dependence is a power dependence.

## 1. Introduction

The technical conditions of the existing line constructions fail to meet the modern requirements in terms of operational performance. The causes of this condition are (1) overloading of the structure, (2) incorrectly designed road structure, and (3) low quality of materials. In many cases, these conditions are associated with the subsoil of structures, structural load-bearing capacity, incorrect design, and inadequately performed quality control of the earth structure during its construction. The failures resulting from an insufficient load-bearing capacity of the subsoil are highly dangerous and require urgent and expensive repairs or reconstructions. Currently, there are many methods, based on various theoretical approaches, for obtaining data on the carrying capacity of the subsoil. To ensure consistency between the different methods, it is necessary to establish conditions for measurements that precisely perform comparative measurements using selected methods (static and dynamic plate load tests). The obtained values were compared to determine the correlation dependence. In general, the quality of the subsoil or pavement is one of the most important factors influencing the life of a road structure.

### 1.1. State of the Art in the World

Today, it is necessary to address research focused on the diagnostic and measurement of the mechanical properties of road pavements. Across the globe, many research teams have dealt with research in this area.

The beginning of the research was based on a single-layer model of the subsoil that was defined by Boussinesq [1] summarised for road pavements. Later, they were derived for application in other simplified approaches: scientists Yoder, Witczak [2], and Burmister [3]. However, modern designs in use today use numerical approaches based on FEM. The first simplified numerical models for layered half-space were applied by Duncan [4], Helwany [5], Park [6]. Nonlinear analysis and the initial use of 3D elements have been intensively studied by Saad [7], Ranadive [8], Rahman [9]. Currently, research is focused on the synthesis of experiments and numerical modelling. To make it possible to describe the realistic behaviour of the subsoil road pavement and its layers (interaction, redistribution, delamination, etc.), it is extremely important to obtain data from experimental tests.

To improve the transport service of every country at the required level, it is necessary to set up appropriate infrastructure, including roads [10,11] and railways [12,13]. The basis of all transport constructions is, to a dominant extent, the subsoil formed by soils. In order for the subsoil of linear transport structures to be able to meet its mechanical properties in a required quality for a long time, it must be built in a reasonable quality at a reasonable price. In Central Europe, quality adequacy is most often checked through static and dynamic elastic modules. The experimental measurements focused on static and dynamic plate load tests (PLT) of subsoil and road pavement subbase layers were presented by Zednik [14], Tompai [15], Elhakim et al. [16], Volovski et al. [17], Abulkareem [18], and Lehmann et al. [19].

Dynamic plate tests are designed to control the quality of earthworks in road and airport constructions which are assumed to have a long, linear, and areal development (typically embankments and road/railway subgrades or road subbase). Generally, the abovementioned studies presume that the use of reworked materials, selected in quarries, and the conditions of compaction (Proctor [20]) are subject to pre-qualification and acceptance by the road engineer, with a target density in situ optimum at a given water content. The construction process is always presumed to provide a small test field to calibrate the compaction energy of the used equipment, such as the number of passes, as well as the thickness of bulk materials that will be subject to compaction.

At this stage, the association of an in-situ test with a dynamic plate allows a comparison of the optimum conditions that have been designed with the value of the dynamic deformation modulus. Therefore, rapid tests are generally useful for site management and the inspection of earthworks, so as to obtain confirmation of the homogeneity of the execution of the works.

The static modulus of rocks is usually different from the corresponding dynamic modulus. The ratio between them is generally complex and depends on several conditions, including stress state and stress history [21,22]. Different drainage conditions, dispersion, heterogeneities and strain amplitude, are all potential reasons for this discrepancy. To a large extent, each mechanism can be expressed mathematically with reasonable precision, thus quantitative relations between the modulus can be established. This provides useful tools for the analyses and prediction of rock behaviour. For instance, such relations may be used to predict static stiffness and even strength based on dynamic measurements. This is particularly useful in field situations where only dynamic data are available [23].

The methods for determining the deformation modulus of unbound pavement materials do vary considerably, and therefore, in the literature, the terms resilient modulus (in labo triaxial), modulus of elasticity, and FWD (in situ) deformation are used to describe the results of such tests. The parameters of the resilient modulus for various types of subgrade soil are influenced especially by loading frequency [21], stress level [24], compaction degree, and moisture content [25]. Authors in [26] present an empirical logarithmic predictive model for the dynamic resilient modulus established based on the static resilient modulus and CBR for cement- and lime-stabilised soils. The paper [27] presents an evaluation of the performance of unbound materials models in predicting the resilient modulus of base layer aggregates using a repeated load triaxial test database and involving gravel/crushed gravel and crushed stone. While the modulus of elasticity is stress divided by strain for a slowly applied load, the resilient modulus is stress divided by strain for rapidly applied loads—such as those experienced by pavements. Knowing the dependences of static and dynamic modulus of elasticity of the subsoil is also important for the quality control of innovative materials in engineering structures of interest [11,28]. Based on the analysis of static and dynamic characteristics of granulated rubber-sand mixtures, as a new type of subgrade filler in railway engineering, with different rubber content, the optimum granulated rubber content should be approximately 10% [29].

### 1.2. Research Background of the Study

The work performed in this article is aimed at the correlation by finding an interpretation key, both theoretical, by introducing the theory of impact and experimental tests with some correlations of real data and subsequent numerical modelling. However, this approach is too simplistic because it is a net correlation without specialising it on batches of measurements that can have other parameters of comparison (density or compaction level, humidity, uniformity in terms of classification, the thickness of the layer of soil being compacted, deformation resistance of soil subbase, etc.).

This article mainly focuses on the road construction sector; however, the applications of modern dynamic plate load tester extend not only to the top of road trench excavations, but also in the body of the embankment and its top (pavement subgrade), in unbound granular mixed foundations (pavement subbase), cement stabilization, and cemented mixtures.

The main goal of this study is to implement the knowledge obtained from the research on subsoil and road pavement conducted in Slovakia. The unique experimental results obtained from the measurements conducted in the research lead to an improvement in the design process of the road. Therefore, it is crucial to collect long-term data from the measurements so that the correlation of individual trials opens into the relevant results usable for the proposal with respect to the FE models.

This paper presents the comprehensive knowledge obtained from the research, which included multiple years of monitoring of deformation characteristics and the degree of compaction of earth structures and base pavement layers, conducted by team Decky et al. [30] and partly mentioned in older publications [31,32]. A systematic approach must be applied to the quality evaluation of earthworks or other civil engineering structures. The synthesis of existing domestic and foreign knowledge, standard procedures, and objectified research results of the authors create a theoretical basis for a systematic approach to quality evaluation of earthworks in civil engineering structures. Part of the research is also to determine the influence of the size of the load plate in the static load test on the course and results of the measurement and to determine the relevant conversion relationships. Partial results were published in [33] and are constantly updated and implemented during the revision of STN 73 6190 in 2019.

### 1.3. Research Contribution of the Study

The results of the long-term research referred to in this article can be divided into three main areas based on national technical standards and regulations:Determination of the required parameters of the load-bearing capacity of earth structures materials and underlays of pavements,The effect of the load plate size in the static load test and the correlation using different load plate sizes,Correlations of the load-bearing capacity determination and the degree of compaction of earth structures and underlays by the direct method (static load test) and the indirect method (dynamic load test).

All testing methods that are typically used to detect the degree of compaction parameters of the soil and unbound pavement layer are listed in Table 1.

The Czech Republic applies to the evaluation of deformation characteristics and degree of compaction of soils and aggregate materials where the required values of deformation modulus and basic correlations between static and dynamic load test. This correlation is comparable to the long-term monitoring conducted by the team of authors.

Earthworks should be considered as structural systems that are an integral part of any civil engineering project (road, railway, sewer, water-mains, etc.). During this period of global economic recession and the time of intensive preparation and construction of the highway infrastructure in the Slovak Republic, the practical implementation of the above knowledge could significantly contribute to the optimization of resources [34,35].

The economic and ecological aspects result in the development of methods and technological procedures for the utilization of the materials with the highest suitability to the earth structures, thus eliminating the need to store unsuitable materials [19,36,37]. From the perspective of the effective use of the quality control time of earth structures, the need to know the relevant correlation dependencies of the static and dynamic load is equally important [20,38,39]. A significant amount of time can be saved if the dynamic load test can be used in a reliable manner. Moreover, the static load test (30–60 min) and dynamic load test (3–5 min) correspond to the price for the dynamic load test, which represents about 20–25% of the price of the static load test [40,41,42].

Multiple numerical simulations have been used to support experimental tests in general. Chapter 5 presents a numerical study of the layered model of the static load test situation and its sensitivity to changes in the one-layer stiffness.

A flow chart illustrating the research process of the problem is shown in Figure 1. One possible suggestion for scientific discussion regarding further research could be considered as this conceptual proposal.

The purpose of the study is to develop credible analytical and numerical models to explain the apparent disproportions in the ratios of static and dynamic deformation models used in evaluating the quality of the mechanical performance of transportation structures. The static load test (SLT) and dynamic load test (DLT) are performed with lightweight loading devices in Central European (CE) conditions for these purposes. There are many situations in which SLT is not feasible due to time, space, or financial limitations, so the conversions of the DLT results to SLT values, which are crucial for the quality control of earth structures, are fundamental. CE uses linear calculations of the above deformation characteristics, aiming to verify their universal validity for all soils.

## 2. Methods—Static Theory of Impact

In terms of the methodology of obtaining the results for the research objectives presented in this paper, it is necessary to define the relationships between the observed variables with respect to the basic theoretical knowledge used in practice. The following subsections are structured to introduce the theoretical background of the correlation relationships between static and dynamic load test results.

### 2.1. Engineering Theory of Impact

The static theory of impact can be considered an approximate impact theory [30]. For the purposes of engineering practice, further simplified assumptions, which have no significant effect on the course of the impact, are introduced. A summary of such solutions is usually designated as the engineering impact theory. For the determination of the dynamic elasticity modulus of the subgrade, the contact between a solid cylinder and an elastic half-space is essential [43,44,45,46].

When a cylinder is pushed into the elastic half-space subject to the contact surface being constantly parallel to the surface plane of the half-space, the following relation is applied between the pressure force *P* (perpendicular to the contact surface) and pushing force *y*:(1)$$P=\frac{2\cdot a\cdot E_{1}}{1-\nu^{2}}\cdot y=C_{1}\cdot y,$$
where: *a*—radius of the cylinder [m]; *E*_1_—elasticity modulus of half-space material [MPa]; *ν*—Poisson’s ratio.

Figure 2 shows the contact between a rigid cylinder and an elastic half-space.
(2)$$-m_{0}\cdot \frac{d^{2}y\left(t\right)}{dt^{2}}-C_{1}\cdot y\left(t\right)=0,$$

After modification:(3)$$\frac{d^{2}y\left(t\right)}{dt^{2}}+\omega_{0}^{2}\cdot y\left(t\right)=0,$$
where:(4)$$C_{1}=\frac{2\cdot a\cdot E_{1}}{1-\nu^{2}},$$
(5)$$\omega_{0}=\sqrt{\frac{2\cdot a\cdot E_{1}}{m_{0}\cdot \left(1-\nu^{2}\right)}}=\sqrt{\frac{C_{1}}{m_{0}}}$$

The entire impact represents half of the vibration and takes place in half of the vibration period *T*_0_. Hence, the time of impact *t_r_* equals.
(6)$$t_{r}=\frac{T_{0}}{2}=\frac{\pi }{\omega_{0}}=\pi \sqrt{m_{0}-\frac{1-\nu^{2}}{2\cdot a\cdot E_{1}}}=\pi \sqrt{\frac{m_{0}}{C_{1}}}$$

Maximum push *v_max_* is calculated as follows:(7)$$v_{max}={\frac{c_{0}}{\omega_{0}}}_{}=c_{0}\cdot \sqrt{m_{0}\cdot \frac{1-\nu^{2}}{2\cdot a\cdot E_{1}}}{=}_{}c_{0}\cdot \sqrt{\frac{m_{0}}{C_{1}}}={\sqrt{\frac{m_{0}\cdot g\cdot h_{0}\cdot \left(1-\nu^{2}\right)}{a\cdot E_{1}}}}_{}$$

Maximum impact force *P_max_* is determined by the following equation:(8)$$P_{max}=C_{1}\cdot y_{max}\cdot ={\sqrt{m_{0}\cdot C_{1}}}_{}=c_{0}\cdot \sqrt{\frac{2\cdot m_{0}\cdot a\cdot E_{1}}{1-\nu^{2}}}={\sqrt{\frac{4\cdot m_{0}\cdot g\cdot h_{0}\cdot a\cdot E_{1}}{1-\nu^{2}}}}_{}$$

The following variables are used in Equations (6)–(8):

*ω*_0_—angle frequency of movement ω=2π⋅v/L [rad·s^−1^];

*c*_0_—velocity of the rigid body impact [m·s^−1^];

*h*_0_—height of the fall of the rigid body [m];

*m*_0_—weight of the rigid body [kg]; and

*g*—acceleration of gravity [m·s^−2^].

The velocity of the rigid body at the impact on the elastic half-space is determined according to Equation (9):(9)$$c_{0}=\sqrt{2\cdot g\cdot h_{0}}$$

### 2.2. Applying the Engineering Theory of Impact in the Calculation of Deformation Characteristics

The engineering theory of impact was applied in the calculation of the deformation modulus [47,48], which is detected by the light dynamic plate of the LDD 100 device (light falling weight deflectometer) (Figure 3).

The basic equation for calculating the dynamic deformation modulus *E_vd_* according to the Slovak Technical Standard STN 73 6192 is:(10)$$M_{r}=\frac{\pi \cdot d\cdot \sigma }{4y_{m1}}\cdot \left(1-\nu^{2}\right)$$

In the instructions for the use of the LDD 100 device, the relation for calculating the impact deformation modulus is stated as follows:(11)$$E_{vd}=\frac{F}{d\cdot y_{e1}}\cdot \left(1-\nu^{2}\right)$$

According to the amended CSN 73 6192 (Czech Technical Standard) of 1996, the dynamic deformation modulus *M_rz_* for subgrade and earthworks fills is calculated as follows:(12)$$M_{rz}=1.57\frac{a\cdot \sigma }{y_{c}}\cdot \left(1-\nu^{2}\right)$$

The following variables are used in Formulas (10)–(12):

*a*—radius of the loading plate [m];

*d*—diameter of the loading plate [m];

*F*—impact force [N];

*σ*—contact stress [Pa];

*y_m1_*, *y_e1_*, *y_c_*—amplitude of deflection at the centre of the loading plate [m];

Although Equations (10)–(12) have equivalent results, the terminological inconsistency that prevails in this area should also be considered. Figure 4 shows the dependence of the maximum dynamic force on the subgrade elasticity modulus (elastic half-space, with Poisson’s ratio 0.35) at an impact of a 10 kg steel cylinder with a 30 cm diameter, from a height of 75 cm.

As an example, Figure 4 highlights the value *P_max_* at the impact of a rigid body onto the subgrade, with an elasticity modulus of 60 MPa and Poisson’s ratio of 0.35. The values of *P_max_* were calculated using Equation (8).
$$P_{max}={\dot{{\sqrt{\frac{4\times 10\;kg\times 9.81\;m\cdot s^{-2}\times 0.75\;m\times 0.15\;m\times 6\times 10^{7}\left(\frac{kg\cdot m\cdot s^{-2}}{m^{2}}\right)}{1-0.35^{2}}=54.9\;kN}}^{}}}_{}$$

The above input values used for the calculation of *P_max_* correspond to the recommended and calibrated values stated for the LDD 100 device, with the following numerical characteristics:▪measured impact deformation modulus: optimum *E_vd_* = 10 to 125 MPa,▪measured deviation: 0.1 to 3.0 mm,▪measuring plate with an installed sensor with a diameter of 300 mm,▪weight of sinker: 10 kg ± 100 g,▪falling height: 750 ± 10 mm,

Applying the relations 6 to 8 for the LDD 100, excluding the influence of shock absorbers for the value of the elasticity modulus of subgrade 1 MPa and *ν* = 0.35, the following is obtained:▪amplitude of the impact force *P_max_* = 7.09 kN▪time of impact *t_r_ =* 0.017 s▪maximum push *v_max_* = 0.0207 m

The values *P_max_* and *t_r_* represent numerical characteristics stated by the manufacturer as the quasi-constant values of the impact impulse and the time of impact.

## 3. Correlations of Static and Dynamic Deformation Characteristics

### 3.1. Correlations of Static and Dynamic Deformation Moduli of Earthworks

Figure 5 presents the objectified power correlation dependency of results for more than 150 pair measurements for the static load test (SLT), and the corresponding dynamic load test performed by the LDD 100 device from the years 1995–2010 [22]. These methods are the most widely used to control the quality of compaction in earthworks and in improving the quality of foundation soil.

The values of *E_vd_* were determined according to Equation (11) and values *E_def,_*_2_ (second loading cycle for the stress range 100 to 200 kPa) according to the relation (13), in compliance with relevant stipulations of STN 73 6190.
(13)$$E_{def}=\frac{\pi }{2}\left(1-\nu^{2}\right)r\frac{\Delta p}{\Delta h}$$
where: 

*E_def_*—deformation modulus in kPa or MPa;

*r*—radius of the plate in metres;

∆*p*—change in contact stress in kPa or MPa;

∆*h*—change in the settlement of the plate at stress change in meters.

The correlation coefficient ranges from −1 to 1, a relatively high value of the correlation coefficient means that there is a high dependency between the variables. This does not necessarily mean that there is also a high causal dependency. The degree of causal dependence is expressed by the coefficient of determination R^2^, a key output in regression analysis. It is the square of the correlation coefficient between variables based on the sample values and it gives valid results when the observations are observed correctly without measurement errors. All correlation dependences in Figure 5 and Figure 6 achieve a correlation coefficient higher than 0.84, which in terms of Spearman correlation coefficient classification represents a very strong correlation degree. Stated correlation coefficients correspond to values of coefficient determinations R^2^ in levels 0.8655; 0.7179 and 0.8427.

Figure 6 presents the objectified correlation dependency in a linear shape, with different materials and levels of earthworks (EW). Measurements of deformation characteristics were performed on various materials of earthworks, and at different structural layers of roads, especially on:▪loamy soils,▪sand and gravel soils,▪mixed soils,▪gravel sand pavement protection layers,▪crushed stones pavement sub-base layers, and▪stony fill pavement sub-base layers.

Figure 7 presents a comparison between our objectified correlation dependencies and the values used in Germany and the Czech Republic [22].

### 3.2. Justification of the Results

It is evident that the equations for the impact of a rigid body on an elastic half-space cannot be fully applied to the dynamic load testing performed by the LDD 100 device; however, they should have general validity. However, from the abovementioned, it can be anticipated that the contact stress under the plate of the LDD100 device will not be identical in all cases. This logical physical premise is supported by the fact that the value of the coefficient *k_SLT/LDD100_* increases with the quality of a material in terms of its deformation characteristics. A better-quality material has to have higher values of elasticity modulus, which generates higher amplitudes of the impact force, which, in turn, exceeds the contact stress, which is stated as constant by the manufacturer.
(14)$$k_{SLT/LDD100}=\frac{E_{DEF,2}}{E_{VD}}$$

According to Equation (8), with the increasing value of the elasticity modulus of the subgrade, the maximum amplitude of the impact force *P_max_* ‘*increases proportionately*’, and the entailing value of *E_vd_*. Figure 8 shows that the coefficient *k_SLT/LDD100_* (relation 14) is 1.0 for modulus value *E_def,_*_2_ = 24.5 MPa, which corresponds to the value of elasticity modulus *E* = 27 MPa.

It follows that for *E* = 27 MPa, approximately identical deformation characteristics are found by static and dynamic load testing. For the falling height of 0.75 m, which is a constant value used in technical practice, the LDD 100 device generates a contact pressure of 0.1 MPa under the plate, especially for the value *E* = 27 MPa. Comparison of values *k_SLT/LDD100_ = E_def,_*_2_/*E_vd_* obtained from the research activities of the authors’ labelled monograph [22]—power dependency (see Figure 5 and Figure 8), with theoretical values obtained according to the static theory of impulse (see Figure 4).

The values of *k_SLT/LDD100_* in the case of the static theory of impact were determined according to the following relation:(15)$$k_{SLT/LDD100}=\frac{E_{DEF,2}}{E_{VD}}\frac{P_{Max,E_{i}}}{P_{Max,E=27MPa}}$$ where *P_Max, Ei_* is the maximum amplitude of the impact force for a particular value of elasticity modulus of subgrade *E_i_* [N], and *P_Ma_*_x, E=27MPa_ is the maximum amplitude of the impact force for the value of elasticity modulus of subgrade *E_i_* = 27 MPa in [N], (see Figure 3).

Based on the abovementioned evidence, it can be unequivocally stated that the range of coefficients *k_SLT/LDD100_* from 0.5 to 4.0 (see Figure 8) is caused by the fact that the LDD 100 device does not generate a constant impact impulse for a constant falling height and different earthworks.

For the approximate determination of the static deformation modulus *E_def,_*_2_ on the measured dynamic deformation modulus *E_VD_*, obtained by LDD apparatus can be used according to [22,49] relation presented in following Table 2. Values in the table were obtained from the German Transport Research Institute on the basis of extensive comparative measurements and evaluation.

## 4. Effect of Load Plate Size on Result of Static Load

This section presents the objectified results in Table 3 of the field research for unstabilised construction pavements (FR UNCOPA), where the alluvial soil and the layer of the R-material (reclaimed asphalt containing alluvial and gravel parts) with an average thickness of 10–15 cm and currently serves as a parking plot for passenger cars extending the original area to form part of the local communications [11,33].

Static load test (SLT) measurements were made on two dates (Figure 9).

▪ load plate (LP) with diameter d = 357 mm, A = 0.100 m^2^,▪ load plate with d = 505 mm and A = 0.200 m^2^,▪ load plate with d = 600 mm and A = 0.283 m^2^.

Based on the analysis of the obtained results, it can be stated that in the case of the quasi-elastic half-space model, the size of the load plate does not have a pronounced influence on the evaluated deformation characteristics. In the opinion of the authors of the recommendation STN 73 6190 on the dimensions of the load plate, in the case of the quasi-elastic half-space model, it can be fully accepted.

## 5. Numerical Simulation of the Static Plate Load Test

Considering the different methods of calculating stresses and deformations in the pavement layers, it is necessary to test the use of numerical models and calculations. Due to the fact that the SLT provides real deformation values, it is possible to model the test situation with FEM. Consequently, this model can be further validated and used for pavement design.

In general, numerical subsoil models can be created using a wide range of different methods. Currently, the most commonly used method is FEM. Different types of FEM models are suitable for different application purposes, including the calculation of static and dynamic responses through their shapes, boundary conditions, types of finite elements, types of contacts, etc. By examining the analytical and experimental results that were presented, the FEM model was supposed to investigate the validity of its use in practice. The material parameters of the presented model are set according to the experiments in the paper. When material properties are verified, it is possible to extend the geometry of a model, add additional materials, and then conduct comprehensive analyses.

The numerical simulation of the load test is based on the results of the numerical model. This model was created using the visual FEA system [50,51]. The computing system used for the simulation was developed for static, dynamic, and other special engineering simulations.

In this software, the finite element method (FEM) is the core method used for numerical computations. A numerical model of the load test was created from standard 3D volume elements hexahedron and prism types. The modelled area had dimensions of 10 × 10 m with a square plan shape. The depth of the layers was 20 cm and 5 m. Layer 1 parameters were as follows: modulus of elasticity *E*_1_ = 350 MPa; Poisson’s ratio *ν*_1_ = 0.3; and unit mass *ρ*_1_ = 2050 kg/m^3^. As shown in Figure 10, the colour of Layer 1 is blue. Modelled Layer 2 parameters are: *E*_2_ = 25–120 MPa, Poisson’s ratio *ν*_2_ = 0.35, and unit mass *ρ*_2_ = 2050 kg/m^3^. As shown in Figure 10, the colour of Layer 2 is green. The connection between Layer 1 and 2 allows the model to be modelled with a special FEM element “Surface Interface”. This element can account for the shear flow in the geotechnical simulations.

One of the most important parts of the FEM model is the steel load plate, which is modelled with volume-meshed elements. These volume parameters are *E_S_* = 210 GPa, Poisson’s ratio *ν_S_* = 0.33, and unit mass *ρ_s_* = 7850 kg/m^3^. The steel circular plate was loaded with a simulated load on the upper surface with a value of *q* = 350 kPa. The diameter of this plate is *d* = 0.375 m, and it has a grey colour (Figure 11). The FEM model contained 3196 volume elements.

The main results obtained from the simulation were the maximal vertical displacement on the steel load plate surface *w_max_* [mm] and the maximal normal stress on the upper surface of Layer 2 *σ*_1,*max*_ [Pa]. Owing to their comparability with experimental tests, these parameters are the most important. Figure 10 presents the contour graphical outputs of the results for *E*_2_ = 25 MPa (part A) and *E*_2_ = 120 MPa (part B).

## 6. Discussion

Using the performed study and obtained research results, correlation dependencies between static and dynamic load tests were determined, detailed analyses of the results, and a comparison with the results of other authors was performed (Figure 12).

An analysis of the differences between the determined correlation and the standard correlations used in Central Europe was conducted. The FEM model was linked to an analytical model using elastic impact theory.
(16)$$E_{def,2,\;Kvalitest}=0.942E_{vd}^{1.176}$$
(17)$$E_{def,2,\;Uniza}=0.380E_{vd}^{1.379}$$

The authors consider the contribution of the article especially in the field of scientific explanation of the apparent disproportions of the correlation coefficients k for various soil subsoil roads and railways and implementation into technical practice. In the field of transfer of scientific knowledge into practice, these correlation dependencies have been implemented in Slovak conditions [52]. The authors allow themselves to present a comparison of their results with the research results of the Polish author Wyroslak. The paper [10] presents site comparative tests based on the light falling weight deflectometer, the static plate load tester, the dynamic probing light tester and the bearing ratio tester (CBR in-situ) with relationships between soil state parameters. Presented correlations between dynamic modulus and secondary static modulus (Equation (18)) were performed by analysing the results obtained by the same devices as presented in this article under boundary conditions very close to the Slovak standards.

The power function describes the relationship between the dynamic modulus and the secondary static modulus:(18)$$E_{vd}=5.1\cdot E_{v2}^{0.48}$$

Coefficient of determination *R*^2^ = 0.9 indicates that 90% of changes in the amount of the dynamic modulus value are explained by the value of the secondary static modulus. The variable *E_v_*_2_ in Equation (18) is identical to the authors presented by the deformation modulus *E_def,_*_2_. For the purposes of credibility comparison, it is necessary to express Equations (19) and (20) from Equations (16) and (17), and Equation (21) from [52].
(19)$$E_{vd}={\left(\frac{E_{def,2,Uniza}}{0.38}\right)}^{\left(\frac{1}{1.379}\right)}=2.01\cdot E_{def,2Uniza}^{0.72}$$
(20)$$E_{vd}={\left(\frac{E_{def,2,Kvalitest}}{0.942}\right)}^{\left(\frac{1}{1.176}\right)}=1.06\cdot E_{def,2Kvalitest}^{0.88}$$
(21)$$E_{vd}={\left(\frac{E_{def,2,TP004}}{1.143}\right)}^{\left(\frac{1}{1.663}\right)}=6.99\cdot E_{def,2,TP004}^{0.60}$$

According to [19] the correlation *E_v_*_2_ versus *E_vd_* is common in German engineering practice, Equation (22) shows the linear correlation proposed in FGSV:2009 (Forschungsgesellschaft für Straßen-Directive on earthworks in road construction) and for gravel in Equation (23) were performed at Technische Universität Darmstadt, Germany [53]. The next conversion between static and dynamic load-bearing capacity moduli is in [15]. The overview of linear and nonlinear (power and logarithmic) relationships between the measured static and dynamic modulus of elasticity was published in [54] (see Figure 13).

For a detailed course of the detected correlations between static and dynamic modulus of elasticity for all soils and rocks and graphical comparison with the works of Central European authors (see Figure 14).
(22)$$E_{vd}=\frac{E_{v2}}{2}$$
(23)$$E_{vd}=\frac{E_{v2}}{3}$$

By comparing the correlation dependencies between the static and dynamic modulus of elasticity obtained by the research in the study (Equation (17)) with the correlation dependencies standardly used in the Central European region (Equation (22)), the following differences and trends can be observed. When converting *E_def,_*_2_ in the range of 5–120 MPa to *E_vd_* according to Equation (17) compared to Equation (22) when sampling 1 MPa, on average it shows values 1.44 times higher. Within each interval, this ratio represents the following values: for 5–30 MPa it is 1.97; for 31–60 MPa it is 1.45; for 61–90 MPa it is 1.25; for 91–120 MPa it is 1.14. In the reverse conversion, converting *E_vd_* to *E_def,_*_2_ 5–30 MPa is 0.5; 31–60 MPa is 0.80; 61–90 MPa is 0.98.

Numerical modelling and FEM simulations can be useful for supplementing knowledge regarding the behaviour of road structure layers. This procedure is important for load testing [55,56]. Numerical results and their correlation with experimental values are crucial components in the road structure design. In this study, the results presented in the numerical study showed relevance in numerical modelling and indicated a good degree of compliance with the tested values. Similar studies based on FEM simulations are presented in [57,58,59].

According to a study from 2019 [21], the quantitative relationship between dynamic and static resilient modulus of subgrade soil has yet to be established. As a consequence, the present research paper focuses on this issue, a culmination of 15 years of research activities of the authors in the area of objectivation of the credibility correlation of the aforementioned modules. Results of tests conducted in the EU conditions for testing the quality of subgrade soil compaction were the subject of the research. According to the TRL (Technology Readiness Levels) classification, the presented research results can be classified as TRL 5 in the EU—technology validated in the relevant environment [60,61,62].

## 7. Conclusions

Based on the analysis of the results of the comparison of static and dynamic load tests, analytical and numerical models of the subsoil formed by the earth and uncemented structural materials, respectively, the universal validity of the linear calculation according to FGSV:2009 [53] was not confirmed. An analysis of the subsoil and an analytical and FEM model demonstrate that the above dependence is better characterized by the power dependence found by the authors. Since in Slovakia, in the context of the Road Act, the design of pavements is carried out according to the applicable standards, technical regulations, and objectively determined research results, it is possible to transfer the acquired knowledge into technical practice almost immediately. The authors have already used their preliminary findings in the development of TP 004 [52] and created conditions for optimising the use of public resources for road pavements.

Based on research performed by a group of authors, which formed the basis for this study, the following recommendations can be stated with regard to earthworks quality control:Systematic approaches should be applied for quality inspection of earthworks;Compaction test results should be used to prepare the earthworks quality control plan when inspecting larger works.When inspecting the degree of compaction in cohesive soils using indirect methods, the actual moisture content of the earthwork that is being assessed must always be checked.To check the load-bearing capacity and the degree of compaction of the earthworks, the static load testing method should be preferred.As specified in the design, static load testing can be applied only if the assessed earthwork has a moisture content within the permissible amount.If the above requirement is not met, the usage of static load testing is subject to the existence of a relevant correlation between the monitored deformation characteristic and moisture.If it is not possible to use static load testing in relation to the LDD 100 device, the estimated recalculation coefficients according to Equation (17) should be used.

Acceptance of the above recommendations would also contribute to better conformity with the requirements of the relevant dimensioning methods for asphalt and concrete pavements under conditions prevailing in the Slovak Republic.

All correlations presented were developed for the subsoil considered as an elastic half-space. In the continuing research in this area, the authors intend to focus on the layered half-space and to create conditions for rapid determination of the deformation characteristics of the subgrade system directly entering into the design of cement concrete pavements.

## Figures and Tables

**Figure 1 materials-15-02922-f001:**
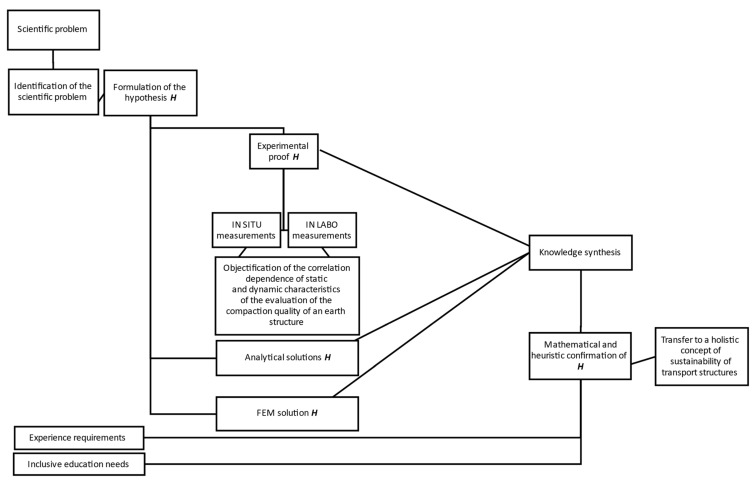
Diagram of investigation related to dynamic vs. static deformation modulus of different materials used for earthworks quality control.

**Figure 2 materials-15-02922-f002:**
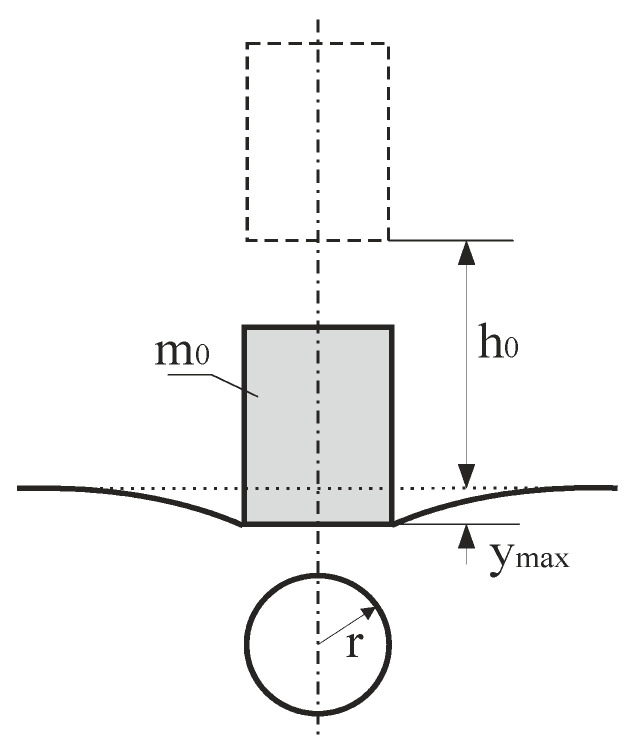
Contact between a rigid cylinder and an elastic half-space.

**Figure 3 materials-15-02922-f003:**
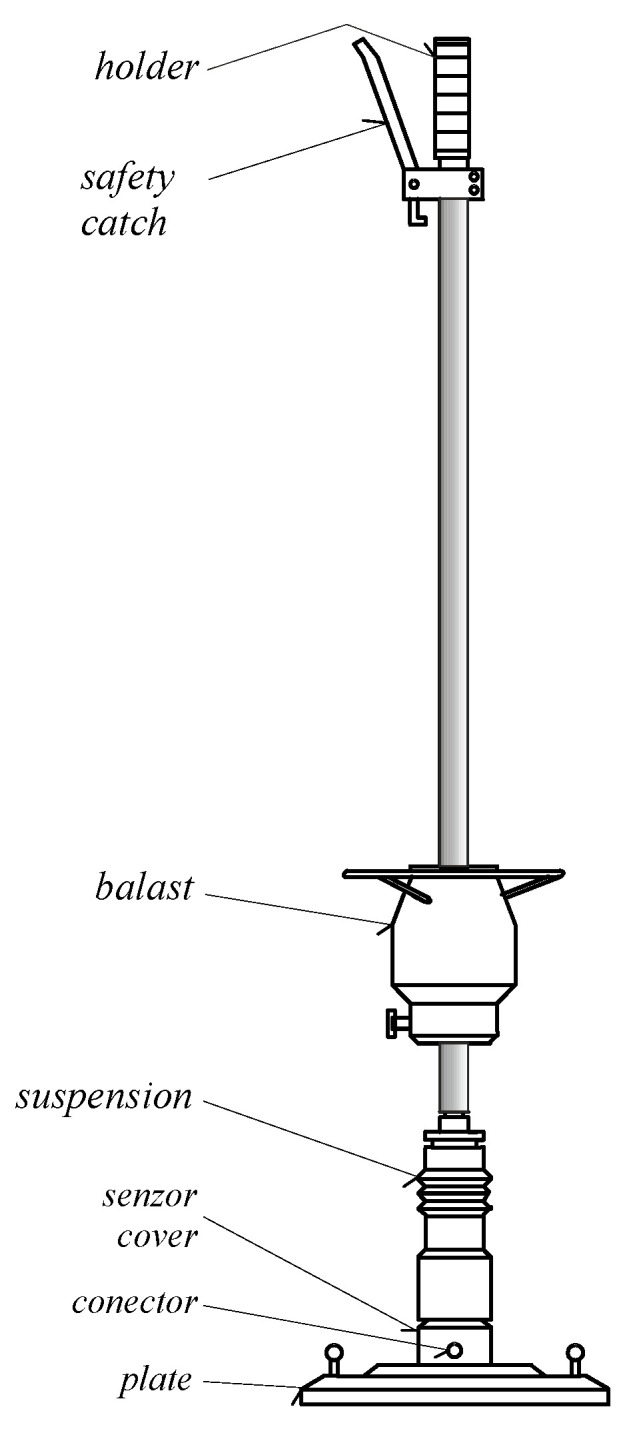
Lightweight Dynamic Plate Testing device LDD 100.

**Figure 4 materials-15-02922-f004:**
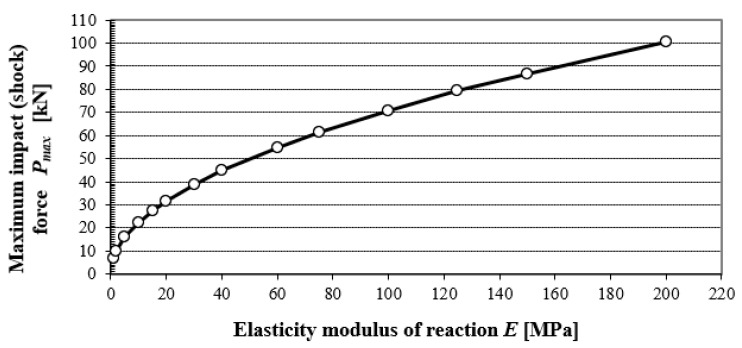
Dependency of maximum impact force on elasticity modulus of reaction of subgrade.

**Figure 5 materials-15-02922-f005:**
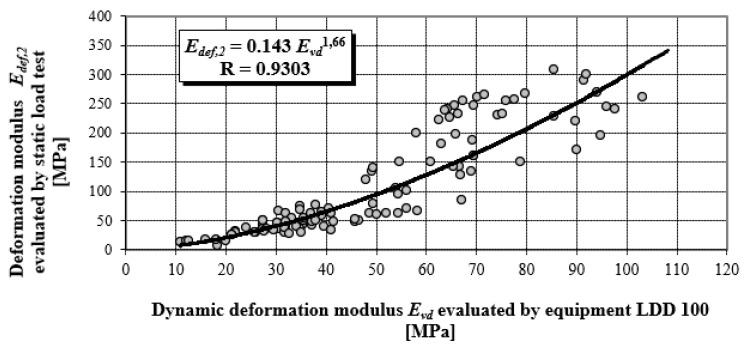
Power correlation dependency of *E_def,_*_2_ on *E_vd_* during the years 1995–2010.

**Figure 6 materials-15-02922-f006:**
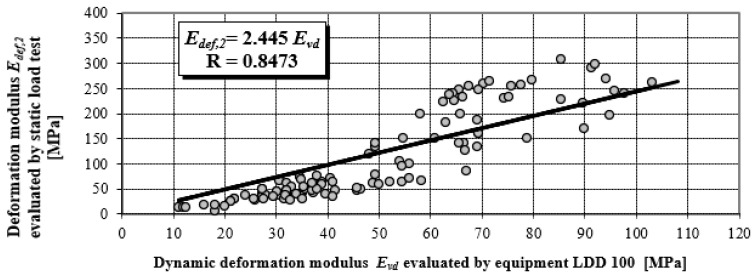
Linear correlation dependency of *E_def,_*_2_ on *E_vd_*.

**Figure 7 materials-15-02922-f007:**
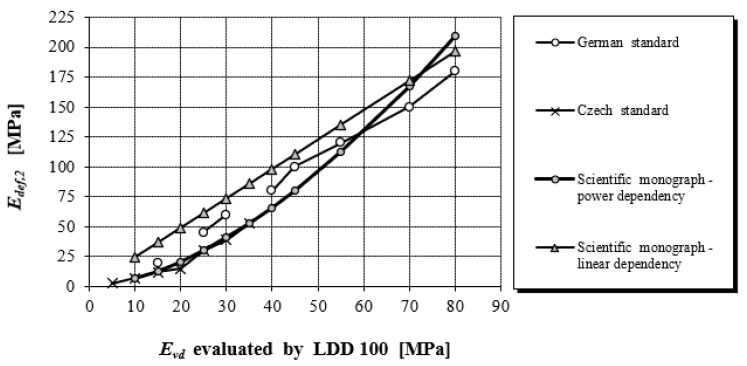
Comparison of objectified correlation dependencies with German and Czech conversion values.

**Figure 8 materials-15-02922-f008:**
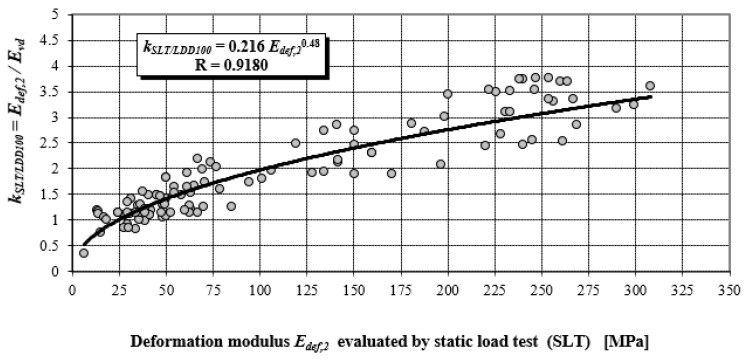
Power correlation dependency of *k_SLT/LDD100_* on *E_def,_*_2_.

**Figure 9 materials-15-02922-f009:**
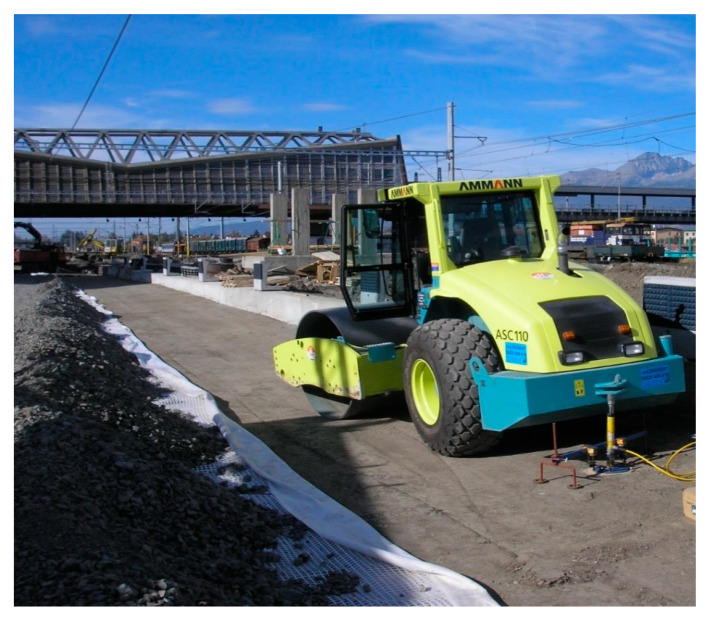
Static load tests on the compacted layer.

**Figure 10 materials-15-02922-f010:**
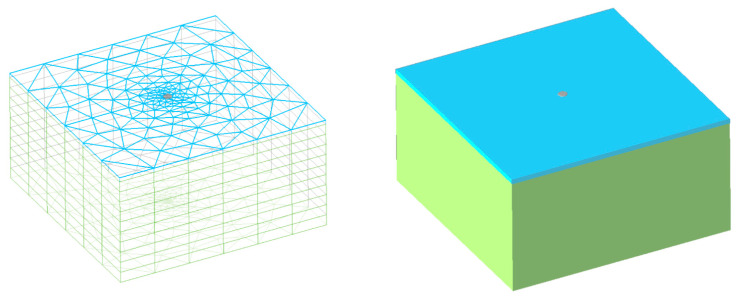
FEM model of the load test with mesh and render.

**Figure 11 materials-15-02922-f011:**
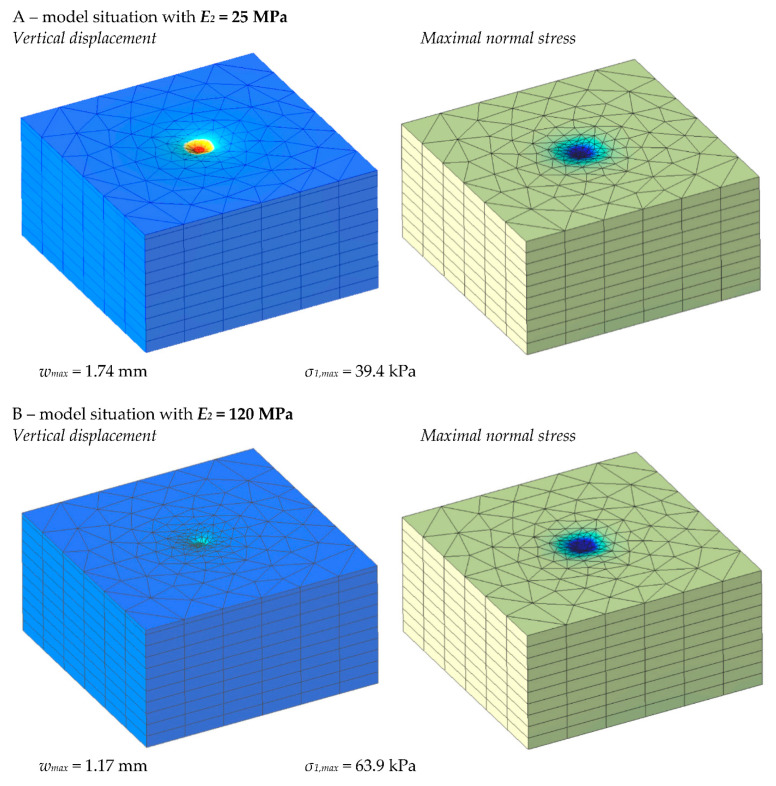
FEM model results for maximal and minimal values *E*_2_.

**Figure 12 materials-15-02922-f012:**
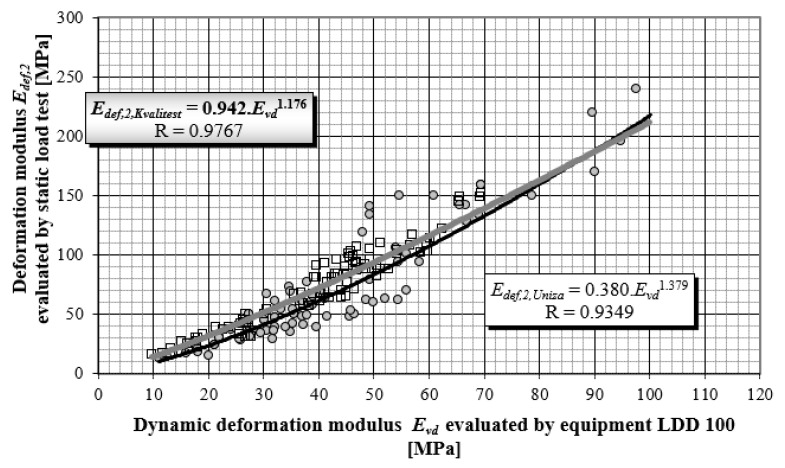
Power correlation dependency of *E_def,_*_2_ on *E_vd_* during the years 1995–2020 by authors from the University of Zilina in cooperation with Kvalitest company.

**Figure 13 materials-15-02922-f013:**
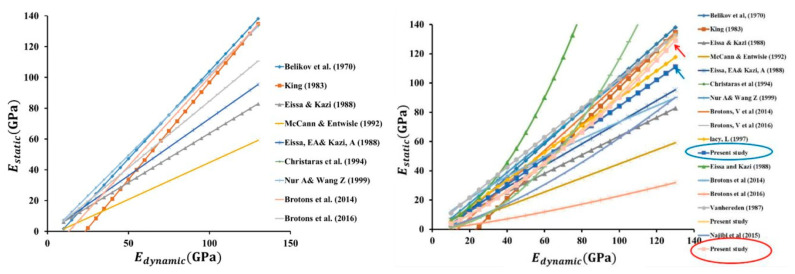
Linear and nonlinear (power and logarithmic) relationships between the measured static and dynamic modulus of elasticity [54].

**Figure 14 materials-15-02922-f014:**
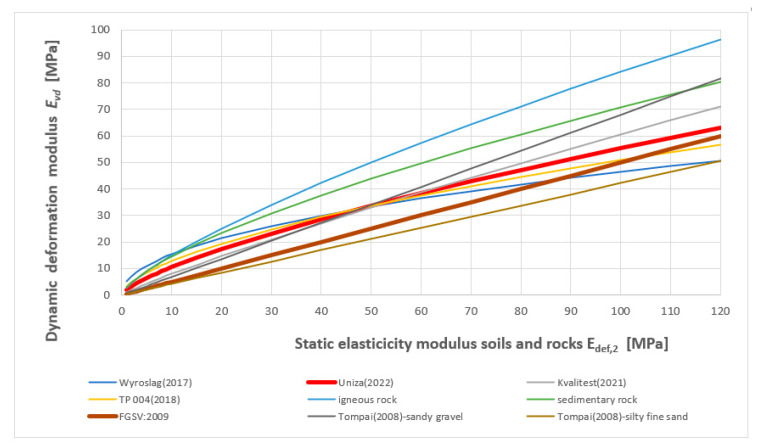
Comparison of authorial relationships between *E_def,_*_2_ and *E_vd_* for all soils and rocks with the works of Central European authors.

**Table 1 materials-15-02922-t001:** Testing methods to detect degree of compaction used in Slovak republic.

Method	Type of Material
Direct methods: Determination of bulk density and humidity, calculation of *D* parameters. static load test	F, S, G, B
Indirect methods: radiometric methods	F, S
dynamic plate load test	F, S, G
geodetic compaction control (levelling)	G, B
dynamic method of compaction control (compaction meter)	F, S, G
penetration tests (static, dynamic)	F, S, G

**Table 2 materials-15-02922-t002:** Recommended relation between *E_vd_* vs. *E_def,_*_2_ used in Germany.

	Relation *E_vd_*/*E_def,_*_2_							
*E_vd_* [MPa]	>15	>25	>30	>40	>45	>55	>70	>80
*E_def,_*_2_ [MPa]	>20	>45	>60	>80	>100	>120	>150	>180

**Table 3 materials-15-02922-t003:** Recapitulation of SLT results with different size of load plate.

Assessed Deformation Characteristics	Average Assessed Deformation Characteristics SLT [MPa] for the Area of the Load Plate [m^2^]								
	A = 0.100 m^2^			A = 0.200 m^2^			A = 0.283 m^2^		
	1. LC	2. LC	Ratio	1. LC	2. LC	Ratio	1. LC	2. LC	Ratio
Modulus of deformation *E_def,i_*	13.8	25.1	1.82	11.9	21.8	1.83	16.8	22.0	1.31

## Data Availability

The data presented in this study are available on request from the corresponding author. At the time the project was carried out, there was no obligation to make the data publicly available.

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
