# Peer review of "The Relationship between Dynamic and Static Deformation Modulus of Unbound Pavement Materials Used for Their Quality Control Methodology"

_materials, 2022, doi:10.3390/ma15082922_

Round 1

Reviewer 1 Report

The authors tried to focus on the relationship between dynamic and static deformation  modulus of unbound pavement materials used for their quality control methodology. Though the authors reported that they specifically deal with the comparison of two commonly used methods, this paper is not suitable for publication in the current form. The abstract of the paper does not highlight its innovation and main research conclusions, which needs to be rewritten. Only the numerical simulation of plate static load test is not enough.  The discussion of this paper is very shallow. The authors should carry out a comprehensive analysis of their data and develop some trends from the observations. Is it appropriate to quote more references in the research conclusions? The conclusions of the paper need to be improved.

Author Response

Dear Reviewer,

Thank you very much for all your comments on our revised manuscript. Below we have provided detailed answers to your comments. We hope you will accept our new layout of the revised manuscript. All introduced corrections were listed and justified in the attached file as suggested by the Editor.

Reviewer 2 Report

SUMMARY

The article is written on a current topic. In modern construction, it is necessary to achieve a theoretical and practical correspondence between planning and control of quality parameters in earthworks. The problem of the study is expressed in the fact that failures that occur due to insufficient bearing capacity of the soil are very dangerous and require urgent and expensive repair or reconstruction. The authors assessed the current state of the issue. The relevant specifications and rules are based on an objective correlation that ensures the correct compatibility between the methods and characteristics used for quality control. The authors present the corresponding correlation dependence of the most commonly used quantitative characteristics in quality control of earthworks. The article has a high practical and scientific potential. She makes a good impression. The authors have done a lot of work. Theoretical premises are backed up by in-depth studies. The article has a strong mathematical apparatus and a calculation part. However, the article has some shortcomings that should be corrected. They are listed below. 

COMMENTS

  1. "Abstract" in its current form does not reflect the problem being solved in the study. It is necessary to add 1-2 sentences about the problem being solved at the beginning.
  2. Section 1 analyzes in detail the works of other authors, but I would like to see an analysis of another 5-10 sources on the research topic for the period 2017-2022 in order to more broadly reflect the current state of the issue.
  3. Probably, a separate section "Methods" should be singled out from the end of section 1. 
  4. Section 1 should be completed with a statement of the purpose and objectives of the study.
  5. Smoother transitions between sections and subsections should be provided.
  6. At the beginning of section 2, the paragraph on lines 152-157 requires a short preamble.
  7. The introduction of additional visualization elements would probably add to the attractiveness of the article. It would probably be worth adding the research program in flowchart format. A similar flowchart is available at the very end of the article. Moving it to the beginning would probably make the article more structured.
  8. It is necessary to correct the numbering of equations: after equation (12), equation (12) follows again.
  9. In Formula (12), after Figure 3, there is no need to give detailed calculations of the parameter Pmax and duplicate Formula (8). It suffices to describe the calculation algorithm and the previously given Formula (8). It would be necessary to optimize the indicated fragment of the article.
  10. It would be necessary to remove the repetition of explanations of formula parameters, for example, the parameter ν is explained several times. If this is the same parameter, then it is enough to explain it once. If these are different parameters, then their designations should be different, for example, ν1, ν2, ν3, and the like. Needs to be corrected or explained.
  11. The first mention of Figure 5 should have come before Figure 5, not after.
  12. Table 2 is followed by Table 4. It is necessary to correct the numbering of the tables.
  13. There is no reference in the text to Table 4. Must be added.
  14. Starting from Figure 9 and up to Figure 12, it is necessary to check the references to them, in some cases they are completely absent, or are already after the figures. Also, erroneous references are sometimes given, for example, in line 420, perhaps there should be a reference to Figure 11, and not Figure 9. Needs to be corrected.
  15. In Figure 11, it is necessary to correct the separator of the fractional part: “,” replace with “.”.
  16. In line 444, references in square brackets should have been given in exactly the same way as in line 452, that is, [42-44].
  17. References to figures in the text are in two different styles: Fig. 1 and Figure 12. It is necessary to bring to a single style in accordance with the design rules approved in the Journal. 
  18. A discussion section should be added. A more detailed comparison of the obtained results with the results of other authors is needed. What are the fundamental differences between the results obtained by the authors and those previously known? In part, this has already been done by the authors in the "Conclusion" section. These sections need to be structured.
  19. In general, it would be more common to see the article structure in the recommended "IMRAD" format. However, this is at the discretion of the authors.
  20. The conclusion is very succinct. In conclusion, it is necessary to more widely disclose and reflect a specific scientific result. It is necessary to add 2-3 paragraphs about the significance for fundamental and applied science, as well as identify promising areas for further research on the topic of quality control of road surfaces.
  21. It is methodologically not entirely correct to cite Figures 11 and 12 in the conclusion. It would probably be worth mentioning them in the previous section.
  22. The list of references should be supplemented with 5-10 fresh sources for the period 2017-2022.
  23. A little check on the style of the English language should be done. 

Author Response

(The authors gave the same response as above.)

Round 2

Reviewer 1 Report

The authors have made great efforts to improve the quality of this manuscript. The revised manuscript has been improved by considering my comments. However, the suggestion is: Minor Revision

(1) The figure 5, 6, 8 should be checked. The meaning of R in these figures shall be explained. R2 is a measure of how well the fit function follows the trend in the data.

(2) Line 350: Change  “Fig. 3” to “Figure 3”.

(3) Line 353: Change  “Fig. 8” to “Figure 8”.

(4) Line 486: Change  “Fig. 13” to “Figure 13”.

(4) Change  “Fig.  ” to “Figure ” .

Author Response

Dear Reviewer,

All of the author team would like to thank you very much for your valuable advice and comments regarding the manuscript. The last comment has also been considered and implemented. We are glad that quality experts still review papers of this type worldwide.

Answer to your comments:

(1) The figure 5, 6, 8 should be checked. The meaning of R in these figures shall be explained. Ris a measure of how well the fit function follows the trend in the data.

- supplemented, explained (Line 303 – 312)

(2) Line 350: Change  “Fig. 3” to “Figure 3”.

- changed

(3) Line 353: Change  “Fig. 8” to “Figure 8”.

- changed

(4) Line 486: Change  “Fig. 13” to “Figure 13”.

- changed

(4) Change  “Fig.  ” to “Figure ” .

- checked and changed